# Severe Neurological Manifestation in a Child with Multisystem Inflammatory Syndrome

**DOI:** 10.3390/children9111653

**Published:** 2022-10-29

**Authors:** Mônica de Oliveira Santos, Diuly Caroline Ribeiro, Jordanna Sousa Rocha, Sibely Braga Santos Maia, André Luís Elias Moreira, Paulo Alex Neves Silva, Célia Regina Malveste Ito, Lilian Carla Carneiro, Melissa Ameloti Gomes Avelino

**Affiliations:** 1Clinical Pathology and Medicine, Federal University of Goiás, Goiânia 74690-900, GO, Brazil; 2Department of Pediatrics, Hospital das Clinicas, Federal University of Goiás, Goiânia 74690-900, GO, Brazil; 3Department of Pediatrics, Hospital Estadual de Doenças Tropicais, Goiânia 74850-400, GO, Brazil; 4Department of Microbiology, Federal University of Goiás, Goiânia 74690-900, GO, Brazil; 5Department of Pediatrics, Federal University of Goiás, Goiânia 74690-900, GO, Brazil

**Keywords:** children, ischemic stroke, SARS-CoV-2

## Abstract

Background and objectives: During the COVID-19 pandemic, we followed with concern the evolution of several children diagnosed with Multisystem Inflammatory Syndrome in Children (MIS-C). The purpose of this study is to describe the evolution of MIS-C in a previously healthy 3-year-old girl. Methods: We tracked the daily medical report of all children admitted with suspected MIS-C to the five largest regional hospitals. Results: Our screening identified a child who had several neurological complications associated with MIS-C. We report hematological alterations, transient cardiac dysfunction, and cerebral involvements such as laminar cortical necrosis caused by ischemic stroke. We present the course of treatment and clinical outcome, and other complications such as a severe subglottic stenosis occurring after extubation. Conclusion: Subglottic stenosis is an expected complication after prolonged intubation, and the presence of dysphonia and/or stridor is an important predictive factor. MIS-C with severe neurological alteration may occur in a healthy child, and early diagnosis and treatment with a pulse of corticoid with immunoglobulin are essential for a favorable outcome.

## 1. Introduction

In April 2020, during the most critical period of the COVID-19 pandemic in England, cases of a new syndrome with varied manifestations of signs and symptoms were reported in pediatric patients infected with SARS-CoV-2. This syndrome was termed Multisystem Inflammatory Syndrome in Children (MIS-C), and had characteristics similar to Kawasaki Disease, incomplete Kawasaki Disease, toxic shock syndrome, and macrophage activation syndrome [1,2].

Although MIS-C can be severe, it is relatively uncommon, with an estimated incidence of 2 per 100,000 individuals less than 21 years old [2]. MIS-C may begin weeks after a child is infected with SARS-CoV-2 [3,4]. It is known that, among other symptoms, most MIS-C cases manifest with non-purulent conjunctivitis, acute gastrointestinal signs and/or symptoms, shock, and cardiac dysfunction. In addition, elevated inflammatory markers and a positive laboratory test for SARS-CoV-2, or a history of contact with a positive case, are considered, according to WHO criteria [1,2].

Published guidelines for the treatment of MIS-C support the use of intravenous immunoglobulin (IVIG) and/or high-dose corticosteroids as the mainstay of first-line therapy [2,3,4].

Our objective was to report a case of MIS-C in a healthy 3-year-old child who developed a very severe neurological condition, presented several changes in hematological parameters, and developed severe subglottic stenosis three weeks after she was discharged from the hospital.

## 2. Case Report

A clinical case of a previously healthy Caucasian female patient aged 3 years and 8 months was admitted to the emergency care unit with fever starting three days before admission. After two days, she evolved with vomiting, prostration, convulsive crises, and lowered level of consciousness, requiring orotracheal intubation.

Due to suspicion of meningitis associated with septic shock, she was referred to the pediatric intensive care unit (PICU). On admission, the patient was in a very serious condition, with signs of shock refractory to initial measures, dehydration, hypothermia, and hypoglycemia. Exchange of the orotracheal tube was necessary on admission, and the patient was placed under neuroprotection despite the cerebrospinal fluid (CSF) showing a subtle increase in cellularity. The patient required continuous infusion of vasoactive amines. Ceftriaxone and acyclovir were also started, due to the suspicion of sepsis and CSF features associated with the possibility of viral encephalitis, respectively.

During evolution, laboratory tests were performed that showed a multisystem inflammatory process. Hematological parameters showed an increase in leukocytes, ferritin, fibrinogen, C-reactive protein (CRP), procalcitonin (PCT), lactate dehydrogenase (LDH), and D-dimer levels. Inflammatory tests of MIS-C and serology for COVID-19 were collected, with positive IgG. Other serological tests (EBV, Varicella zoster, Herpes virus 1 and 2, Parvovirus) were negative; bacterial and fungal cultures were also negative. Treatment was performed with pulse therapy of corticosteroids and intravenous immunoglobulin (IVIG) (2 cycles), the first being 2 g/kg and the second 1 g/kg, showing significant improvement.

A computed tomography (CT) scan of the head showed recent frontoparietal-occipital ischemia, resulting from laminar cortical necrosis related to a subacute chronology (Figure 1A). A magnetic resonance (MR) scan showed cortical necrosis in the acute/subacute phase with extensive area of gliosis of the adjacent encephalic and cerebellar parenchyma, involving the frontal, parietal, and occipital lobes, the posterior ends of the temporals with greater extension to the left, and the cerebellar hemispheres (Figure 1B). Chest CT showed segmental atelectasis in the upper right and lower left lung lobes (Figure 1C).

The echocardiogram showed signs of pancarditis, mild systolic dysfunction of the left ventricle, laminar pericardial effusion, and mild dilation of the left main coronary artery and the proximal portion of the right coronary artery. However, ten days after a second echocardiogram showed all parameters within normal limits.

The patient required invasive mechanical ventilation for 12 days (cannula with cuff). An unsuccessful extubation attempt occurred within 7 days of admission. In the second extubation attempt, intermittent non-invasive ventilation (NIV) was initiated interspersed with a high-flow catheter, with good results. The patient presented post-extubation expiratory stridor, which was controlled by nebulization with epinephrine and dexamethasone. The patient also had elevated D-dimer levers (27,561 ng/mL). Intravenous corticosteroid therapy (35 mg/kg/day) was maintained according to the hospital’s MIS-C protocol, in addition to anticoagulants such as acetylsalicylic acid (5 mg/kg/day).

On the eleventh day of admission to the PICU, the use of enoxaparin 2 mg/kg/day was initiated but was followed by thrombophlebitis in the right upper limb. During hospitalization, the patient developed difficult-to-control anemia, requiring four blood transfusions. In an expanded survey with hematology, searches were performed for primary antiphospholipid syndrome (APS), autoantibodies, and thrombophilias. The replacement of folic acid and elemental iron at a dose of 4 mg/kg was also started.

The patient presented neurological/motor sequelae resulting from the ischemic condition, with right hemiparesis predominantly in the upper limb and aphasia. She evolved, however, with neurological and motor improvement.

The patient remained in the PICU for 33 days, evolving with clinical and laboratory improvements, and her D-dimer levels decreased (1032 ng/mL). In the infirmary, she became active, obeying simple commands and verbalizing punctual words, with some improvement in hoarseness. She continued to have right hemiparesis with progressive improvement, and she was walking with support. She also accepted oral diet well, and had a good tolerance for the nasoenteric tube.

In her evolution, she also presented ischemic skin lesions in the head pole, and pressure lesions in the gluteal and dorsal regions. The lesions underwent daily dressing changes, with chemical and mechanical debridement when necessary, evolving with gradual improvement and with no signs of secondary infection. Vision in both eyes was affected after the neurological incident (light stimulus was not followed), and ophthalmologic examination found no changes in the eyes.

In total the patient was hospitalized for 40 days, with all blood cultures and pathogens tested being negative. At discharge, the patient was without stridor but with hoarseness, and with right hemiparesis, supported locomotion, impaired vision in both eyes, and progressive signs of improvement in all affected organs.

Two weeks after hospital discharge, she presented progressive discomfort, with the presence of stridor, wishbone and costal retraction, and nose wing beat. The patient was admitted to the emergency room for an airway endoscopy procedure, Myer–Cotton grade III stenosis (90% obstruction) was observed, and she underwent balloon laryngoplasty. After two weeks, a new laryngeal dilatation procedure was necessary with infiltration of triamcinolone, and dilatation was repeated after another three weeks.

The patient recovered with normal breathing at home after three balloon laryngoplasty procedures, and presented motor and visual difficulties with signs of slow improvement.

## 3. Discussion

After the SARS-CoV-2 pandemic, MIS-C is a new clinical manifestation in children and adolescents. It has features of a systemic inflammatory response such as Kawasaki Disease and toxic shock syndrome [2]. MIS-C occurs in patients aged 0 to 21 years who progress to hospitalization and/or death, and is related to the following characteristics: fever observed for three or more days; disease involvement in two or more organs; laboratory evidence indicating increased inflammation; evidence of SARS-CoV-2 infection (history of laboratory contact or positive); and negative tests for other causes of infectious origin [1].

Several studies have been published in recent months demonstrating clinical and laboratory changes in patients with MIS-C. The most predominant symptoms are fever, gastrointestinal symptoms, cardiac abnormalities, and changes in hematological parameters [3,4,5,6,7]. Alkan et al. [8] show specific routine laboratory test results that may be useful in determining MIS-C disease severity, possibly predicting the prognosis and allowing the early initiation of appropriate treatment. Our patient had increased laboratory parameters of leukocytes, CRP, ferritin, D-dimer, LDH, and fibrinogen, described for cases of MIS-C [8,9].

Acute ischemic stroke is a rare emergency in children. The lack of validated and evidence-based data on thrombolytic and endovascular treatments in children represents the main limiting factor for the prevention of life-threatening or disabling sequelae associated with acute ischemic stroke [10].

Several articles have reported mild, moderate, and severe neurological manifestations in children and adolescents with MIS-C. Lindan et al. [11] reported seven cases of stroke, characterized as thromboembolic or vasculitic. In an international case series Beslow et al. [12] reported eight pediatric stroke patients; seven of these cases had acute ischemic stroke, including one newborn, and the other case was a nine-year-old boy with cerebral synovenous thrombosis of the right transverse sinus and right sigmoid sinus. Schupper et al. [13] reported two cases: a five-year-old boy presented with rapid clinical worsening with right middle cerebral artery (MCA) infarction, cerebral edema, diffuse contralateral subarachnoid hemorrhage, and death; and a two-month-old boy with bilateral MCA and posterior cerebral artery (PCA) infarctions with hemorrhagic transformation. In their US case series LaRovere et al. [14] reported 12 cases with acute ischemic or hemorrhagic stroke. Finally, Sánchez-Morales et al. [15] published two cases of acute ischemic stroke in a case series in Mexico. This case report also presents a patient with MIS-C and moderate to severe impairment of the nervous system, in consistency with cases described in the literature.

The risk factors for the development of intubation-related airway complications are well known [16,17]. Our patient developed subglottic stenosis that required to be treated with three balloon laryngoplasty procedures. The patient was discharged from the hospital without stridor but presented with hoarseness; we therefore believe that it is essential to endoscopically investigate all patients with stridor and hoarseness after 72 h post-extubation [16,17].

## 4. Conclusions

Subglottic stenosis is an expected complication after prolonged intubation, and the presence of dysphonia and/or stridor is an important predictive factor. MIS-C with severe neurological alteration may occur in a healthy child, and early diagnosis and treatment with a pulse of corticoid with immunoglobulin are essential for a favorable outcome.

## Figures and Tables

**Figure 1 children-09-01653-f001:**
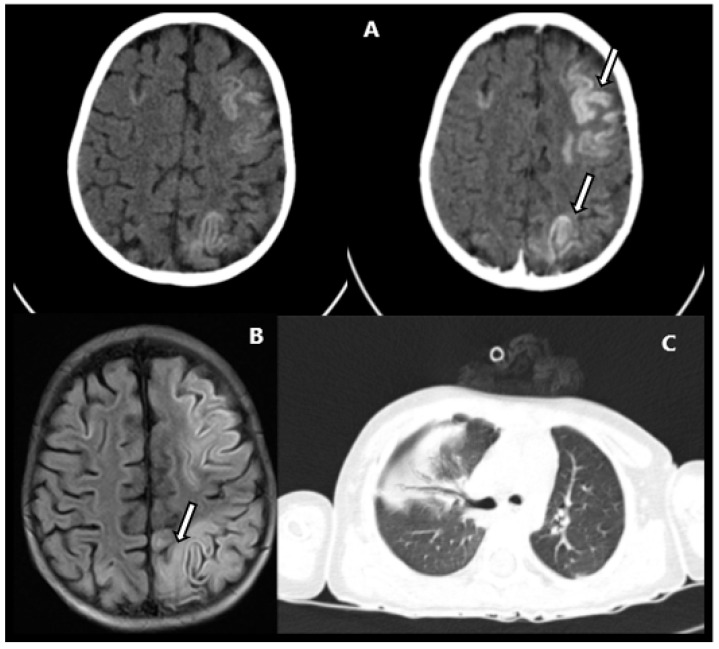
(**A**). CT without (left) and with (right) contrast, showing areas of recent ischemia and laminar cortical necrosis (arrows). (**B**). MR presented cortical HT1 signal in the brain without hemorrhage (arrow). (**C**). Chest CT showing segmental atelectasis in the upper right and lower left lung lobes.

## Data Availability

The data presented in this study are available on request from the corresponding author.

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
