# Peer review of "Severe Neurological Manifestation in a Child with Multisystem Inflammatory Syndrome"

_children, 2022, doi:10.3390/children9111653_

Round 1
Reviewer 1 Report
1. In the keywords, use the full term "Multisystem Inflammatory Syndrome in Children" instead of its abbreviation.
2. "3.8 years old"? This is not a mean age value to be reported in this way. Please provide the age in years and months.
3. Use the full term "cerebrospinal fluid (CSF)" when first mentioned. The same for other abbreviations such as "NIV", "PICU"... etc.
4. Line "89". "He" refers to whom?!! The same in lines 95-96.
5. Line 92-93, please determine the dose of each drug directly after its name to avoid confusion.
6. The case report is a bit confusing. I couldn't follow exactly what was the sequence of the case, symptoms, treatment and the patient response. I think it should be revised and rewritten based on the time sequence.
7. It was not clear whether the MIS-C in this case was secondary to an infection or not? Please explain.
8. What is the point of using "Covid-19" in the keywords? I think it has nothing to do with this particular case.
9. The discussion is so brief and shallow. Only some of the case's symptoms were discussed. What about other symptoms and what about the intervention and the response of the patient?
10. A conclusion is required to wrap up this case study.
11. Language and sentence construction should be revised and improved.
Author Response
Questions:
- In the keywords, use the full term "Multisystem Inflammatory Syndrome in Children" instead of its abbreviation.
The term has been placed
- "3.8 years old"? This is not a mean age value to be reported in this way. Please provide the age in years and months.
Yes, we corrected the age.
- Use the full term "cerebrospinal fluid (CSF)" when first mentioned. The same for other abbreviations such as "NIV", "PICU"... etc.
We searched for all abbreviations and put in the full terms.
- Line "89". "He" refers to whom?!! The same in lines 95-96.
The paraphrase has been rearranged for better understanding.
- Line 92-93, please determine the dose of each drug directly after its name to avoid confusion.
The paraphrase has been rearranged for better understanding.
- The case report is a bit confusing. I couldn't follow exactly what was the sequence of the case, symptoms, treatment and the patient response. I think it should be revised and rewritten based on the time sequence.
Some paragraphs have been rewritten to improve understanding.
- It was not clear whether the MIS-C in this case was secondary to an infection or not? Please explain.
MIS-C occurs after an infection, we have added information to the introduction to clarify the matter.
- What is the point of using "Covid-19" in the keywords? I think it has nothing to do with this particular case.
The literature always brings the term MIS-C accompanied by the statement of contact with the virus and or the development of the Covid-19 disease.
- The discussion is so brief and shallow. Only some of the case's symptoms were discussed. What about other symptoms and what about the intervention and the response of the patient?
We have increased the discussion of the case close to the limit allowed by the scope of the journal. We hope it has improved understanding.
- A conclusion is required to wrap up this case study.
We add a conclusion
- Language and sentence construction should be revised and improved.
Another general text correction was made
Notes:
We appreciate all corrections and accept all possible modifications.
The terms have been revised and the information has been reorganized for greater clarity in the report.
Our focus has always been ischemic stroke, so we chose to change the title of the case report.
We analyzed the references and modified some. Two references were removed and four were added.
We are available for any necessary changes and clarifications.
Yours sincerely

Reviewer 2 Report
This paper reports particular subglottic stenosis, which developed the subglottic stenosis about one month after the intubation. Therefore, the reason for developing subglottic stenosis is unclear. The authors are reporting this patient’s progress since the first day. However, since the cause of subglottic stenosis is unclear, the discussion section should be removed, and all explanations should be moved to the introduction section.
Introduction:
Overall, the introduction needs improvement. The author needs to bring some discussions about other patients in the literature. Did they also have subglottic stenosis? Since the most common reason for subglottic stenosis is the intubation duration, it is essential also to bring their intubation duration.
Also, what is the aim of this report? Because subglottic stenosis is usually the result of intubation. Is it not the case here? Is there any relation between the MIS-C and causing the subglottic stenosis?
Line 36 – 37: Is MIS-C only seen in children? If available, it is better to mention the age range of the children.
Line 37 – 41: Please mention the characteristics.
Line 42: “It’s” => “It is”
Line 42-43: reference?
Case Report:
This section is written well. However, the last two paragraphs should emphasize that the patient didn’t develop subglottic stenosis after 12 days of intubation. Also, if I am correct, she didn’t develop the subglottic stenosis for another 28 days that she was in the hospital. But then she developed it in two weeks.
Also, is there any data showing how much the intubation caused bruises in the subglottic stenosis region?
I saw both “he” and “she” pronouns in this section; I had difficulty understanding if both are referring to the same patient or if it is another person you are referring to. Please make it consistent.
Discussion:
After reading this section, I believe the authors should move most of this section to the introduction section.
As I can see, there is no discussion regarding the reasons for the subglottic stenosis. From my point of view, a particular case is reported for unknown reasons. Therefore, I believe most of the discussions here should be moved to the introduction section and probably removed from the discussion section entirely?!
Author Response
Revisor 2
Este trabalho relata estenose subglótica particular, que desenvolveu a estenose subglótica cerca de um mês após a intubação. Portanto, a razão para o desenvolvimento de estenose subglótica não é clara. Os autores relatam a evolução deste paciente desde o primeiro dia. No entanto, como a causa da estenose subglótica não é clara, a seção de discussão deve ser removida e todas as explicações devem ser movidas para a seção de introdução.
Introdução:
No geral, a introdução precisa de melhorias. O autor precisa trazer algumas discussões sobre outros pacientes na literatura. Eles também tinham estenose subglótica? Uma vez que o motivo mais comum de estenose subglótica é a duração da intubação, é essencial trazer também a duração da intubação.
Also, what is the aim of this report? Because subglottic stenosis is usually the result of intubation. Is it not the case here? Is there any relation between the MIS-C and causing the subglottic stenosis?
yes, we have reviewed the information in the article introduction
Line 36 – 37: Is MIS-C only seen in children? If available, it is better to mention the age range of the children.
yes, we added the characteristics of the disease
Line 37 – 41: Please mention the characteristics.
yes, corrected
Line 42: “It’s” => “It is”
yes, corrected
Line 42-43: reference?
yes, we added the references
Case Report:
This section is written well. However, the last two paragraphs should emphasize that the patient didn’t develop subglottic stenosis after 12 days of intubation. Also, if I am correct, she didn’t develop the subglottic stenosis for another 28 days that she was in the hospital. But then she developed it in two weeks.
Also, is there any data showing how much the intubation caused bruises in the subglottic stenosis region?
I saw both “he” and “she” pronouns in this section; I had difficulty understanding if both are referring to the same patient or if it is another person you are referring to. Please make it consistent.
Discussion:
After reading this section, I believe the authors should move most of this section to the introduction section.
As I can see, there is no discussion regarding the reasons for the subglottic stenosis. From my point of view, a particular case is reported for unknown reasons. Therefore, I believe most of the discussions here should be moved to the introduction section and probably removed from the discussion section entirely?!
Answers
We appreciate all corrections and accept all possible modifications.
Terms have been revised and information has been reorganized for clarity in the article.
MIS-C only occurs in children and adolescents aged 0 to 21 years after contact with Sars-Cov-2. Reported cases over the age of 21 are called MIS-A. In the introduction to the case report, we added information about age and characteristics of the MIS-C.
Our focus has always been ischemic stroke, so we chose to change the title of the case report and emphasize stroke.
Analisamos as referências e modificamos algumas. Duas referências foram removidas e quatro foram adicionadas.
Estamos à disposição para quaisquer alterações e esclarecimentos necessários.
Com os melhores cumprimentos

Round 2
Reviewer 1 Report
The manuscript has been improved. I have only minor comments:
1. Line 72: Please add the unit of the measured IgG (68.5).
2. The language and grammar of the manuscript should be revised.
3. The conclusion should be written as separate sentences. No need to write three unrelated sentences connected with commas and "and".
Author Response
Dear reviewer,
The manuscript was analyzed and revised.
We appreciate all corrections. We are available for any clarification and review.
Yours sincerely

Reviewer 2 Report
The manuscript is now improved a lot. However, again, I am not following the discussion part. It needs to be more clear regarding what are the results of this work. Every time I try to read it, confuses me. Also, regarding the conclusion, I can not see a obvious relation between the discussion part and conclusion. The conclusions need to be addressed completely in the manuscript.
Author Response

(The authors gave the same response as above.)
